# Red-Brown Pigmentation of *Acidipropionibacterium jensenii* Is Tied to Haemolytic Activity and *cyl-*Like Gene Cluster

**DOI:** 10.3390/microorganisms7110512

**Published:** 2019-10-30

**Authors:** Paulina Deptula, Iida Loivamaa, Olli-Pekka Smolander, Pia Laine, Richard J. Roberts, Vieno Piironen, Lars Paulin, Kirsi Savijoki, Petri Auvinen, Pekka Varmanen

**Affiliations:** 1Institute of Biotechnology, University of Helsinki, 00014 Helsinki, Finland; olli-pekka.smolander@helsinki.fi (O.-P.S.); pia.k.laine@helsinki.fi (P.L.); lars.paulin@helsinki.fi (L.P.); petri.auvinen@helsinki.fi (P.A.); 2Department of Food Sciences, University of Copenhagen, Rolighedsvej 26, DK-1958 Frederiksberg C, Denmark; 3Department of Food and Nutrition, University of Helsinki, 00014 Helsinki, Finland; iida.loivamaa@helsinki.fi (I.L.); vieno.piironen@helsinki.fi (V.P.); 4New England Biolabs, Ipswich, MA 01938-2723, USA; roberts@neb.com; 5Division of Pharmaceutical Biosciences, University of Helsinki, 00014 Helsinki, Finland; kirsi.savijoki@helsinki.fi

**Keywords:** genome, haemolysis, granadaene, pigmentation, *Acidipropionibacterium thoenii*, *Acidipropionibacterium virtanenii*

## Abstract

The novel *Acidipropionibacterium* genus encompasses species of industrial importance but also those associated with food spoilage. In particular, *Acidipropionibacterium acidipropionici*, *Acidipropionibacterium thoenii*, and *Acidipropionibacterium jensenii* play an important role in food fermentation, as biopreservatives, or as potential probiotics. Notably, *A. jensenii* and *A. thoenii* can cause brown spot defects in Swiss-type cheeses, which have been tied to the rhamnolipid pigment granadaene. In the pathogenic bacterium *Streptococcus agalactiae,* production of granadaene depends on the presence of a *cyl* gene cluster, an important virulence factor linked with haemolytic activity. Here, we show that the production of granadaene in pigmented *Acidipropionibacterium,* including *A. jensenii, A. thoenii,* and *Acidipropionibacterium virtanenii*, is tied to haemolytic activity and the presence of a *cyl*-like gene cluster. Furthermore, we propose a PCR-based test, which allows pinpointing acidipropionibacteria with the *cyl*-like gene cluster. Finally, we present the first two whole genome sequence analyses of the *A. jensenii* strains as well as testing phenotypic characteristics important for industrial applications. In conclusion, the present study sheds light on potential risks associated with the presence of pigmented *Acidipropionibacterium* strains in food fermentation. In addition, the results presented here provide ground for development of a quick and simple diagnostic test instrumental in avoiding potential negative effects of *Acidipropionibacterium* strains with haemolytic activity on food quality.

## 1. Introduction

*Acidipropionibacterium jensenii* belongs to the newly created genus *Acidipropionibacterium* that originates from the former genus *Propionibacterium*, currently divided into four genera: *Propionibacterium, Acidipropionibacterium, Cutibacterium*, and *Pseudopropionibacterium* [1]. While *A. jensenii* is now the type species of the novel genus, relatively little is known about this bacterium. A draft genome of the type strain *A. jensenii* DSM 20535 (accession number NZ_AUDD00000000.1) and recently a whole genome of its equivalent strain NCTC13652 (accession number LR134473.1) have become available in public databases; yet, a genome-wide identification and description have not been available for the species thus far. 

*A. jensenii* was first isolated from buttermilk by van Niel [2] and the species belonged to the traditional group of “classic” or “dairy” propionibacteria distinguished from the skin-associated, “cutaneous” reclassified corynebacteria [1,3]. While *Propionibacterium freudenreichii* is the main “dairy” species used as a ripening starter in Swiss-type cheese production, *A. jensenii,* among other species, is frequently isolated from milk as well as Swiss and other type of cheeses [4]. In addition to their role in cheese production, the potential applications of dairy propionibacteria include their use as probiotics and biopreservatives in food, as well as the production of biomolecules like vitamins, conjugated linoleic acid, exopolysaccharides, and trehalose [5,6,7,8,9]. Furthermore, *A. jensenii* is one of the organisms utilized in industrial propionic acid production, and research has been conducted to improve the productivity and propionic acid titres [10,11]. In addition, several strains were shown to produce bacteriocins such as Jenseniin G [12], Jenseniin P [13], and Propionicin SM1 [14]. Promising future applications include the use of *A. jensenii* fermentate as a natural antifungal preservative in foods [15]. 

Although “dairy” propionibacteria are not known to carry virulence factors [16], some strains of *A. jensenii* and *Acidipropionibacterium thoenii* were shown to have β-haemolytic activity [17]. β-haemolysis is a virulence property and an undesirable attribute in the safety assessment for food cultures [18]. The haemolytic activity of *A. jensenii* and *A. thoenii* strains has been tied to their red-brown pigmentation [19], which was later shown to result from the production of the polyene pigment granadaene [20]. While the haemolytic activity is a clear food safety concern, pigmented strains of propionibacteria also cause defects in cheeses during ripening by red or brown spot formation [21,22]. Molecular methods based on PCR have been studied to obtain tools for the detection and control of the propionibacteria content in raw milk [23,24]. However, the genetic basis behind haemolytic activity and red-pigmentation has remained unresolved, which hampers development of strategies to eliminate red/brown -spot defects caused by dairy propionibacteria. 

In the pathogenic bacterium *Streptococcus agalactiae,* the production of granadaene and the associated haemolytic activity is an important virulence factor. It is implicated in the bacterium’s ability to efficiently infect human hosts, [25] and was recently shown to enable penetration of the amniotic cavity in pregnant women, which is associated with preterm births and foetal injury [26]. While β-haemolytic activity is a virulence factor that can also indicate potential pathogenicity of the granadaene-producing *Acidipropionibacterium* strains, there are no reports of adverse health effects associated with these strains. Intriguingly, pigmentation in strains of various propionibacteria species has been found to offer protection against growth inhibition by whey filtrates, which was not observed with non-pigmented strains [27]. Thus, it is tempting to speculate that granadaene in these bacteria plays a similar but so far not well understood role as in haemolytic streptococci, which are also resistant to inhibition by milk compounds [28]. 

In *S. agalactiae,* the intensity of pigmentation was shown to correlate with the extent of haemolytic properties, while no haemolysis is observed in non-pigmented strains and no pigmentation in non-haemolytic strains [25]. Since a similar pigmentation related to haemolysis was observed for *A. jensenii* and *A. thoenii* [20], red pigmentation in propionibacteria could be considered a potential indicator of haemolytic activity in these bacteria. However, in the new propionibacteria species *Pseudopropionibacterium rubrum* SK-1^T^ isolated from the human mouth (specifically from gingival sulcus) [29,30] red pigmentation was reported, while no haemolysis on sheep-blood agar was observed. Also, β-haemolysis was recently added to the subspecies description of *Cutibacterium acnes* subsp. *acnes* [31]; yet, no pigmented phenotype has been linked either to the Christie-Atkins-Munch-Petersen (CAMP) factor-mediated co-haemolytic reaction [32] or the β-haemolysis to putative haemolysin *tly* [33]. Thus, there is some controversy in the linking of pigmentation and haemolysis in propionibacteria species and it is not clear whether the phenomenon is restricted to members of the genus *Acidipropionibacterium*. 

In the current work, PCR detection with *cylG* as a target shows diagnostic potential in detection of haemolytic and pigmented strains among several *Acidipropionbacterium* species. In addition, we report whole genome sequences of two novel strains of *A. jensenii* isolated in Finland from barley: JS279 and JS280, which display a distinct pigmented and non-pigmented phenotype, respectively. Here, the sequenced genomes were compared to those defined for the type strain *A. jensenii* DSM 20535 and other acidipropionibacteria. We also complement genomic findings related to B_12_ vitamin synthesis, genome integrity, haemolytic activity and carbohydrate fermentation profiles with necessary bioinformatic and phenotypic analyses. 

## 2. Materials and Methods 

### 2.1. Growth Conditions

Bacterial strains used in this work are listed in Table 1. *A. jensenii* JS279 (VTT E-113203) and *A. jensenii* JS280 (VTT E-113204) were previously isolated from barley samples (data not shown) and deposited in the VTT Culture Collection. The strains were maintained in 15% glycerol stocks at −80 °C and routinely grown on yeast extract-lactate medium (YEL) [34], first for four days on solid medium supplemented with 1.5% agar (Difco, Becton, Dickinson and Company, Sparks, MD, USA) at 30 °C under an anaerobic atmosphere (Anaerocult, Merck, Darmstadt, Germany) and subsequently in 10 mL of liquid medium in 15-mL Falcon tubes for 3 days at 30 °C, under a microaerobic atmosphere.

### 2.2. Isolation of DNA, Genome Sequencing, and Assembly

Genomic DNA was isolated as described previously [36] using Illustra MiniSpin kit (GE Healthcare, Chicago, IL, USA) for JS279 and MagAttract HMW DNA Kit (Qiagen, Germantown, MD, USA) for JS280 samples. The sequencing was performed on PacBio RSII platform with the P5/C3 and P4/C2 chemistry for the strain JS279 and with the P5/C3 chemistry for the strain JS280. The genomes were assembled with Hierarchical Genome Assembly Process version 3 algorithm, with resulting 215X and 380X genome coverage for strains JS279 and JS280, respectively. Base modifications and motifs were detected using the RS Modification and Motif analysis protocol (SMRT Analysis package v.2.3.0). The whole genome sequences of the strains JS279 and JS280 together with the modification profiles were submitted to NCBI for annotation with the Prokaryotic Genome Annotation Pipeline (PGAP) and are available under accession numbers CP025571 and CP025570, respectively.

### 2.3. Bioinformatics Analyses and Classification of the Strains

The average nucleotide identity (ANIb) values were estimated by BLAST algorithm with the aid of JSpecies web software [37]. For the genome-wide descriptions, the genomes of the strains JS279 and JS280 together with the whole genome of *A. jensenii* NCTC13652 (accession number LR134473.1) (equivalent of the type strain *A. jensenii* DSM 20535) downloaded from NCBI were aligned with Progressive Mauve [38], re-annotated with PROKKA [39] and subjected to comparative genomics analysis with Roary [40].

### 2.4. Haemolytic Activity

#### 2.4.1. Detection of Haemolysis and Pigmentation

The haemolytic capability of the strains was assessed by growth on tryptic-soy agar (Becton, Dickinson and Company, Franklin Lakes, NJ, USA) supplemented with 5% sheep blood (Labema, Helsinki, Finland). A fresh colony from a YEL agar plate was streaked on blood agar and incubated for one week at 30 °C under an anaerobic atmosphere (Anaerocult, Merck, Darmstadt, Germany) after which the plates were examined for haemolysis. The presence of clearing around the bacterial growth was interpreted as β-haemolysis. For detection of pigmentation, fresh colonies were streaked on YEL agar plates and incubated as above followed by visual inspection. *Staphylococcus aureus* 8325-4 [41] was used as a positive control of haemolytic activity. The assay was repeated at least three times for each tested strain.

#### 2.4.2. PCR Amplification of the *cylG* Gene Fragment 

Oligonucleotide primers fw 5′ CACCGCGATCGAGATGGC 3′ and rev 5′ GGAACATGTCGGTGTCGACGA 3′ were used for amplification of an internal fragment of the *cylG* gene. The primer annealing to a conserved region in the *cylG* gene, were designed on the basis of BLASTn alignment between the strains *A. jensenii* LMGT 2818 (FJ617193.1), *A. jensenii* JS279 (CP025571), *A. thoenii* DSM 20276^T^ (NZ_KE384018.1), and *A. virtanenii* DSM 106790 ^T^ (CP025198). For crude DNA template preparation, YEL-grown cells were harvested by centrifugation at 3220× *g* for 10 min and lysed by bead beating with glass beads in TE-buffer (10 mM Tris-HCl, 0.1mM EDTA, pH 8.0) with a FastPrep^®^-24 Homogenizer (MP Biomedicals, Irvine, CA, USA) in three cycles of 30 s. Cell debris and glass beads were removed by centrifugation at 12,000× *g* for 5 min and an aliquot of the supernatant was used as a template in PCR. The amplification was carried out using an MJ Mini^®^ Personal Thermal Cycler (Bio-Rad, Hercules, CA, USA) in a reaction volume of 25 μL in Phusion Master Mix HF buffer (Thermo Fisher Scientific, Waltham, MA, USA) including 12.5 pM of fw and rev primers and 1 μL of a crude template DNA preparation. The following thermocycling conditions were used: 98 °C for 2 min, in 30 cycles of 98 °C for 10 s, 70 °C for 15 s, and 72 °C for 15 s. A 5-μL aliquot of the PCR reaction was mixed with 2 μL of loading buffer and run on a 1.0 % agarose gel at 60 V in 1× TBE buffer (89 mM Tris, 89 mM boric acid and 2 mM EDTA) and visualized in an AlphaImager (ProteinSimple, San Jose, CA, USA) device under UV light setting. The sizes of PCR products were determined by comparing against a phage Lambda DNA digested with PstI. The expected size of the *cylG* specific amplicon was 518 bp. Confirmation of the identity of amplified products was performed by Sanger sequencing the amplicons obtained from *A. thoenii* DSM 20276 using BigDye Terminator v3.1 cycle sequencing with the primers fw and rev. Sequences were generated on an Applied Biosystems 3730XL high-throughput capillary sequencer (Thermo Fisher Scientific, Waltham, MA, USA). The resulting sequences were subjected to a BLASTn search of the NCBI database with default parameters [42]. 

### 2.5. Further Characterisation of the Strains JS279 and JS280

#### 2.5.1. Detection of Defence Mechanisms and Mobile Genetic Elements

The genomes of the strains JS279 and JS280 were further subjected to the search of clustered regularly interspaced short palindromic repeats (CRISPR) with the aid of CRISPRFinder [43]. For the search of genomic islands IslandViewer IV [44] and putative prophages Prophinder [45] and Phaster [46] were used. Methylation profiles and Restriction-Modification systems were analysed at REBASE [47].

#### 2.5.2. Phenotypic Characterisation

The carbohydrate fermentation profiles of the strains JS279 and JS280 were determined with the aid of API CHL50 test (Biomerieux, Marcy I’Etoile, France) as described previously [48]. The ability to grow in modified YEL with pH values adjusted to 4.0, 4.5, 5.0 and 9.0, in the presence of salt concentrations adjusted to 256.7 mM (1.5%), 513.3 mM (3%), and 1.1 M (6.5%) NaCl as well as growth in unmodified YEL at 12 °C and 42 °C. These experiments were performed on 96-well microtiter plates (353077, Falcon, Corning incorporated, Durham, NC, USA) as previously described [48].

## 3. Results and Discussion

### 3.1. Classification of Strains JS279 and JS280 as A. jensenii

The results of the ANIb analysis (Figure 1) clearly show that the newly sequenced strains JS279 and JS280 belong to the species *A. jensenii* with the ANIb % identity values ≥97% [37]. The summary details of their genomes are presented in Table 2, together with the details of the previously sequenced type strains of other species belonging to the genus *Acidipropionibacterium*: *A. jensenii* DSM 20535 (NCTC13652), *A. acidipropionici* DSM 4900 (CGMCC1.2230), *A. thoenii* DSM 20276, and *A. virtanenii* DSM 106790 (JS278). In addition, the type strain of the closely related *Propionibacterium freudenreichii* DSM 20271 (JS16) and the most studied strain of *Cutibacterium acnes* DSM 16379 (KPA171202) were included. Genomes of the *Acidipropionibacterium* species are generally larger than those of *Cutibacterium* and *Propionibacterium* representatives, however, *A. jensenii* genomes are smaller than those of other acidipropionibacteria. Interestingly, the type strain *A. jensenii* DSM 20535 has an approximately 5% larger genome than the other two strains and it possesses an additional ribosomal RNA operon. 

### 3.2. Haemolytic Activity, Pigmentation and cylG Gene Detection by PCR

In *S. agalactiae,* the pigment with haemolytic activity is encoded by the *cyl* operon consisting of 10 genes: *cylI, cylF* [51]*, cylE, cylB, cylA, cylZ, acpC, cylG, cylD,* and *cylX* [52]. Later, two additional genes *cylJ* and *cylK* were also identified [53]. While the *S. agalactiae* mutants deficient in either the *acpC*, *cylZ*, *cylA*, *cylB,* or *cylE* gene lost both the pigmentation and the haemolytic activity completely [52], the mutations in *cylF*, *cylI*, *cylJ,* and *cylK* ([52,53] led to milder activities. BLASTp search of the JS279 theoretical proteome against the proteins coded by the *S. agalactiae cyl* cluster gave very low scores, with an average sequence identity of ~30%. However, some of the hits were organised in a cluster and subsequent search against Conserved Domains Database [54] revealed that despite low sequence conservation the functional protein domains belonged to the same families (Figure 2a). BLASTn analyses of available propionibacteria genomes revealed the presence of a putative *cyl* cluster in at least one strain of *A. thoenii* (DSM 20276) and *A. virtanenii* (JS278) (Figure 2b) but not in any of the *A. acidipropionici* strains. The presence of a *cyl* gene cluster was previously reported for a genome assembled from a metagenomic sample, *Propionibacterium* sp. 5 U 42AFAA [26]. The sample is most likely a strain of *Cutibacterium acnes* subsp. *acnes* based on an ANIb value of 99.7% when compared to *Cutibacterium acnes* type strain ATCC 6919. However, as it is impossible to determine the phenotype of this isolate, it was not pursued further.

The subsequent alignment of the putative *cyl* cluster from all acidipropionibacteria showed a high degree of conservation within the *cylG* gene region, which was selected for primer design and subsequent detection of the *cyl* cluster by PCR. Seven available *A. jensenii* strains, two *A. thoenii* strains, and the type strains of the recently described species *A. virtanenii* and that of *P. freudenreichii* as a negative control (see Table 1) were used. In addition, the strains were tested for haemolytic activity and pigment formation on sheep blood plates and YEL plates, respectively. 

Clearance around bacterial growth on blood plates indicative of β-haemolytic activity was observed for five strains including two out of five *A. jensenii* strains (JS279 and DSM29275), the *A. virtanenii* strain, and both *A. thoenii* strains (Figure 3a). Four *A. jensenii* strains and the *P. freudenreichii* strain were judged non-haemolytic on sheep blood plates under the conditions used (Figure 3a). The strains showing β-haemolytic activity also produced red/orange pigment on YEL-plates whereas no pigmentation was observed in non-haemolytic strains (Figure 3b), which is in accordance with previous reports linking haemolytic activity and red pigmentation in *P. jensenii* and *P. thoenii* [20]. Our primers produced amplicons of the expected size (518 bp) from each of the five β-haemolytic and pigmented *Acidipropionibacterium* strains representing three species but not from any of the non-haemolytic and non-pigmented strains (Figure 3c). Sanger sequencing of the PCR product from *A. thoenii* HAMBI 247 and subsequent BLASTn analysis revealed a best hit (97% identity over 499 bp sequence stretch) with *cylG* in the putative granadaene gene cluster of *Propionibacterium jensenii* LMGT2818 (nucleotides 6177–6675 in GenBank entry FJ617193) confirming the specificity of the primers. Thus, PCR detection with *cylG* as a target shows diagnostic potential in detection of haemolytic and pigmented strains among several *Acidipropionibacterium* species. In the present study, we used all the *A. jensenii*, *A. thoenii,* and *A. virtanenii s*trains accessible in DSMZ and HAMBI culture collections. A more thorough assessment of molecular detection will become feasible when more strains are available for research.

About 1–5% of human *S. agalactiae* isolates are non-haemolytic, which is often associated with the loss of activity due to the presence of an insertion sequence in one of the *cyl* genes [25]. Conversely, in *A. jensenii* strains for which genomic sequence is available and no haemolytic activity can be observed, the granadaene gene cluster is absent altogether. This could indicate that in this species haemolysis is a result of acquisition of genes through horizontal gene transfer, which affected only a few of the strains, rather than being a core feature of the species. On the other hand, the widespread occurrence of pigmentation among *A. thoenii* strains could suggest that production of granadaene is intrinsic to this species. However, a larger number of strains is necessary to confirm this notion.

### 3.3. Further Characterisation of the Strains JS279 and JS280

#### 3.3.1. Biosynthesis of Vitamin B_12_

Propionibacteria, particularly *P. freudenreichii,* are known for their ability to produce active vitamin B_12_. However, we previously reported that the members of the genus *Acidipropionibacterium* lack the *bluB* part of the fusion gene *bluB/cobT2* necessary for the production of 5,6-dimethylbenzimidazole (DMBI), the lower ligand of the active form of vitamin B_12_ [35]. Also, the presence of aspartate in the active site of the product of *cobT* gene is decisive for its function. CobT enzyme is responsible for activation of the lower ligand prior to its attachment. The charged amino acid in the active site of acidipropionibacteria indicates preference for activation of lower ligand bases less hydrophobic than DMBI, such as adenine, which results in a preferential production of pseudovitamin B_12_ [55]. A similar organisation of the B_12_ biosynthetic gene clusters can be observed in *A. jensenii* (Figure 4a) as in *A. virtanenii* [35] and *A. acidipropionici* [56], with the exception of the gene clusters I and III being co-located. Accordingly, in our previous report we determined the preference for production of pseudovitamin B_12_ by the strains JS279 and JS280 and that only a small amount of active vitamin is produced if DMBI is provided [57]. 

#### 3.3.2. Mobile Elements and Defence Mechanisms

Analysis with CRISPRfinder identified 14 potential CRISPR arrays, two of which are associated with a complete CRISPR-Cas system (Figure 4b) in each JS279 and JS280 strains. The arrays PossibleCRISPR1 and CRISPR2 of the strain JS279 share a direct repeat (DR) consensus sequence CGGATTACCCCCGCTCGCGCGGGGACGAC. CRISPR2 consists of 76 spacers located between 280463–285125 and is associated with a complete CRISPR-Cas type IE system (Figure 4a). CRISPR1 consists of two spacers and is located directly upstream of the CRISPR2 array (279029–279180). CRISPR1 and CRISPR2 are separated by three genes coding for hypothetical proteins, each with similarity to transposases. Both the organisation of the Cas genes and the DR consensus sequence are highly similar to the CRISPR-Cas system type IE identified in *P. freudenreichii* [48], while DR also is nearly identical to the one associated with the incomplete CRISPR-Cas system type IU identified in *A. virtanenii* (GGGCTCACCCCCGCATACGCGGGGCGGAT) [35]. The first two arrays of the strain JS280, CRISPR1 and CRISPR2 consist of 55 and 15 spacers, respectively, and share the DR sequence GTCGCTCCTCTTTCCAGAGGAGCCCTTCGTTGAGGC. The arrays are separated by a gene coding for ISL3 family transposase (C0Z10_01425) and are located directly upstream of a complete CRISPR-Cas system type IU, which shares high similarity with the corresponding system identified in *P. freudenreichii* [48]. 

The remaining 12 arrays in each strain consist of one to two spacers each and DR sequences 23–43 nt long. While they do not appear to be associated with CRISPR-Cas systems, the presence of DR and their mostly intergenic locations may imply some other, yet unknown role. BLAST search against the spacers did not return hits to any known bacteriophages or mobile elements. Both CRISPR-Cas systems type IE and IU were previously detected in other propionibacteria [35,48], with a similar organisation, but with different DR sequences. It is worth noting that the novel IU system in *A. jensenii* and *A. virtanenii* differs from the one reported in *P. freudenreichii* by the presence of a hypothetical protein, which contains a predicted DNA-binding domain.

Four complete Restriction Modification (RM) systems were identified in both the strain JS279 (Appendix A) and JS280 (Appendix A). The PacBio-aided detection of methylation allowed identification of four 6mA methylation motifs: GAAYNNNNNTGG, CGCATC, GCAGCC and GGAABCG in the strain JS280, the first of which was successfully tied to the activity of the Type I RM system (COZ10_10535-COZ10_10545). No definitive motifs were identified in strain JS279. However, it is worth mentioning that the predicted system (C0Z11_09855, C0Z11_09860, C0Z11_09875) constitutes a part of a complete BREX [58] system (C0Z11_09845- C0Z11_09890).

Nineteen genomic islands were identified for each strain JS279 and JS280, while no prophages were identified in either. Further details of the bioinformatics analyses can be seen in Appendix A (strain JS279) and Appendix A (strain JS280).

Comparison of the genome organisation between the three complete genomes of *A. jensenii* revealed the presence of multiple inversions (Appendix A). A similar phenomenon was reported in *P. freudenreichii* [48], but not in *C. acnes* [59]. Results of comparative genomics analyses between the *A. jensenii* type strain and the two newly sequenced strains can be seen in Appendix A as well as Appendix A.

#### 3.3.3. Phenotypes

Aside from the obvious difference in pigmentation associated with production of granadaene by strain JS279, strains JS279 and JS280 display similar phenotypes (Table 3). The exceptions include a weak growth of strain JS280 already at pH 4.5, while both JS279 and JS280 grew well at pH 5.0. In addition, the strain JS280 was able to grow at 42 °C, while the maximum temperature for growth recorded for strain JS279 was 37 °C. The only difference concerning fermentation profiles was the ability of strain JS280 to utilise salicin as carbon source. More pronounced differences in fermentation profiles can be observed when both strains of cereal origin are compared against the type strain *A. jensenii* DSM 20535 isolated from buttermilk. Both cereal strains are able to grow on methyl-α-D-glucopyranoside, starch, and xylitol, while the dairy strain was able to use D-cellobiose and gentiobiose. The ability to ferment raffinose by the strains of *A. jensenii* and *A. virtanenii* analysed here could be used in the fermentation of pulses. Pulses are considered valuable source of protein where the presence of raffinose is considered one of the antinutritional factors, and currently efforts are made to eliminate it through the use of microbial fermentation [60]. 

It is worth noting that, unlike the type strain *A. virtanenii* JS278, which produced granadaene only when grown on lactate or turanose, strain JS279 invariably produced granadaene also on galactose, inositol, maltose, saccharose, trehalose, melezitose, xylitol, and L-arabitol (data not shown). However, the implications of this difference are currently unknown. 

## 4. Conclusions

Due to their beneficial properties, the dairy propionibacteria including strains of *A. jensenii* and *A. thoenii* have gained interest as secondary and/or protective starter cultures in various food fermentations [7,63,64,65] and even as probiotic candidates [66]. Linking the presence of the *cyl* operon to haemolytic activity and the production of red-pigment allows identification of haemolytic/pigmented strains by whole genome sequencing. Furthermore, it makes it possible to develop specific PCR methodology based on *cylG* gene specific primers. Lack of haemolysis/pigmentation-encoding genes can be used as one of the selection criteria for new propionibacteria strains to be used in food production and/or in search of new probiotics. This will allow to avoid potential negative effects on the food quality or on the health of consumers. While for the moment there are no reports indicating negative health effects of the consumption of cheeses containing pigmented propionibacteria, the presence of haemolytic pigment associated with virulence of the pathogenic *S. agalactiae* raises safety concerns. Until the safety of pigmented propionibacteria is confirmed, the risk of potential adverse effects on consumer health could be avoided by detecting haemolytic strains in raw milk. The strains can then be eliminated by heat treatment from batches intended for production of cheese varieties allowing growth of propionibacteria. 

## Figures and Tables

**Figure 1 microorganisms-07-00512-f001:**
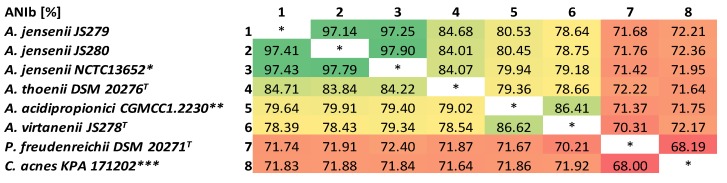
Average nucleotide identities (%) (BLAST-based) between the newly sequenced *A. jensenii* JS279 and *A. jensenii* JS280 strains and type strains of the most closely related species for which the genome sequence is available. * Equivalent of the type strain DSM 20535; ** Equivalent of the type strain DSM 4900; *** Not type, but the most studied strain of *Cutibacterium acnes*; ^T^ indicates type strain of the species.

**Figure 2 microorganisms-07-00512-f002:**
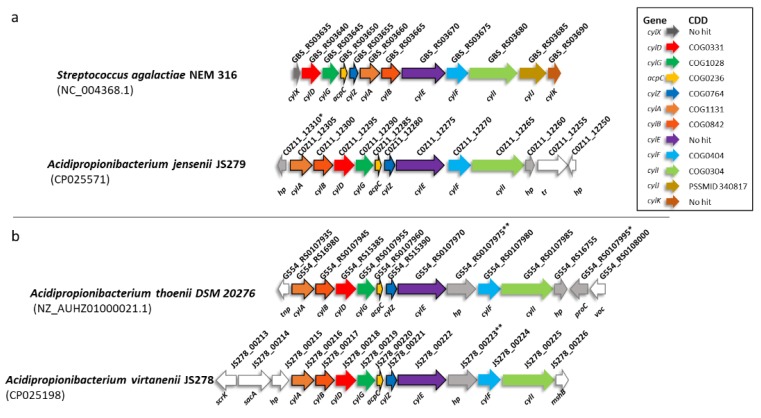
Genes and organization of *cyl* gene cluster in (**a**) *Streptococcus agalactiae* and *A. jensenii* JS279; (**b**) *cyl* gene cluster in other acidipropionibacteria. The genes previously shown as necessary for production of pigment and haemolytic activity in *S. agalactiae* (*acpC, cylZ, cylA, cylB, cylE*) are outlined in black. Homologous genes are marked with matching colours, the conserved domains in gene products are shown in the figure legend. Genes without homology to *S. agalactiae cyl* cluster are light grey. Genes without homology between acidipropionibacteria are additionally filled in white. * Proteins encoded by COZ11_12310 and G554_RS0107995 share 100% sequence identity over 95 aa. ** No homologue is found in the strain JS279 annotated with PGAP. In PROKKA annotation a hypothetical protein encoded by a gene located at 2750821–2752134 shares a 65% identity with JS278_00223.

**Figure 3 microorganisms-07-00512-f003:**
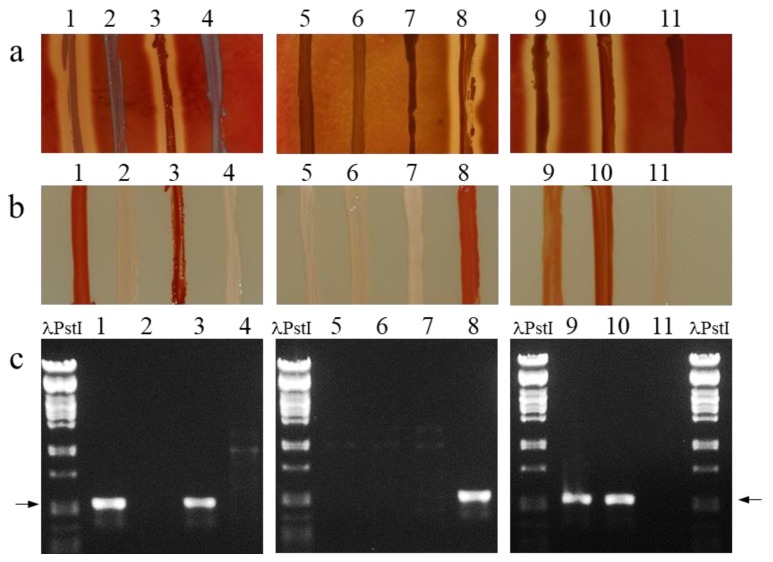
Identification of haemolytic, pigmented and *cylG* carrying strains. Appearance of a streak of bacteria after 7days incubation at 30 °C under anaerobic atmosphere on (**a**) sheep blood agar plate and (**b**) YEL agar plate. (**c**) PCR amplification products of *cylG.* Strains analysed: (1) *A. jensenii* JS279; (2) *A. jensenii* JS280; (3) *A. jensenii* DSM 20275; (4) *A. jensenii* DSM 20535; (5) *A. jensenii* DSM 20278; (6) *A. jensenii* DSM 20279; (7) *A. jensenii* DSM 20274; (8) *A. virtanenii* JS278; (9) *A. thoenii* DSM 20276; (10) *A. thoenii* DSM 20277; (11), *P. freudenreichii* DSM 20271. λPstI refers to phage lambda DNA digested with PstI. Arrows indicate the positions of the 514-bp band in the marker lanes. Representative results of experiments repeated at least three times are shown.

**Figure 4 microorganisms-07-00512-f004:**
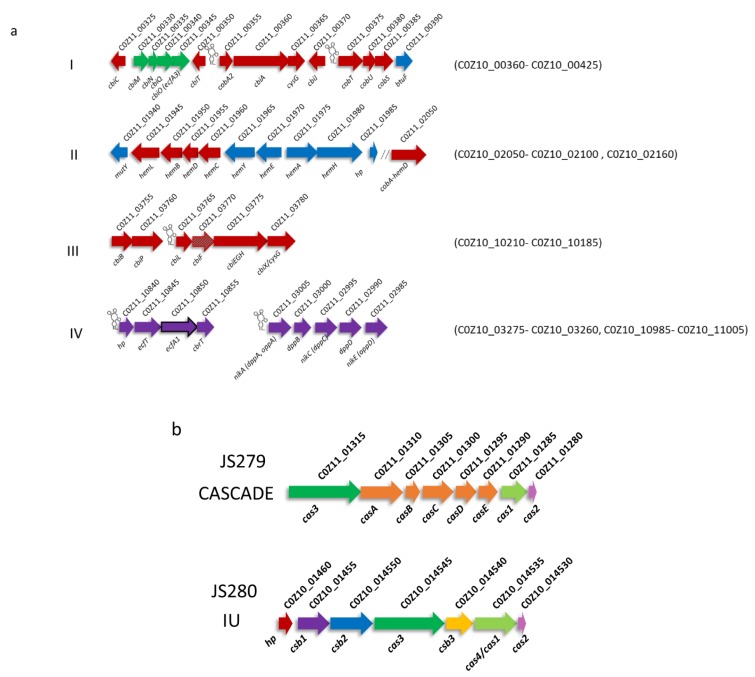
Further characterization of strains JS279 and JS280. (**a**) The four gene clusters of the B_12_ biosynthetic genes identified in strain JS279. Genes directly involved in B_12_ biosynthesis are marked in red. The cobalt transporter genes are marked in green, while the genes involved in the production of porphyrin are marked in blue. B_12_-riboswitch-controlled putative B_12_ (C0Z11_10840-C0Z11_10855) and nickel (C0Z11_02985-C0Z11_03005) transporters are marked in purple. Loci of the corresponding gene clusters in strain JS280 are indicated in brackets. The gene C0Z11_10850 in the putative B_12_ transporter, annotated in JS280 as a pseudogene (C0Z10_03265), is outlined. (**b**) CRISPR-Cas systems Type IE (or CASCADE) in strain JS279, similar to the one previously reported in *P. freudenreichii* as well as a novel Type IU in strain JS280, similar to those previously reported in *P. freudenreichii* as well as *A. virtanenii*.

**Table 1 microorganisms-07-00512-t001:** Bacterial strains used in this study.

Strain	Strain Information	Source/Reference
*Acidipropionibacterium jensenii*		
JS279; VTT E-113203	Isolated from malted barley	VTT Culture Collection
JS280; VTT E-113204	Isolated from malted barley	VTT Culture Collection
DSM 20535	Type strain; Isolated from buttermilk	German collection of microorganisms and cell cultures (DSMZ)
DSM 20275	Isolated from buttermilk	DSMZ
DSM 20278	Isolated from buttermilk	DSMZ
HAMBI 243; DSM 20274		Microbial Domain Biological Resource Centre HAMBI (HAMBI)
HAMBI 245; DSM 20279	Isolated from cheese	HAMBI
*Acidipropionibacterium thoenii*		
HAMBI 247; DSM 20276	Type strain; Isolated from cheese	HAMBI
DSM 20277	Isolated from cheese	DSMZ
*Acidipropionibacterium virtanenii*		
JS278; DSM 106790	Type strain; Isolated from malted barley	[35]
*Propionibacterium freudenreichii*		
DSM 20271	Type strain; Isolated from cheese	DSMZ

**Table 2 microorganisms-07-00512-t002:** Genome summaries of the strains included in ANIb analyses.

Strain	Genome Size (bp)	G+C mol%	Genes	CDS	rRNAs (5S,16S,23S)	tRNAs	Refseq	Genome Status	Reference
*A. jensenii* JS279	3032477	68.6	2672	2610	3,3,3	50	CP025571	complete	(this study)
*A. jensenii* JS280	3044937	68.8	2699	2637	3,3,3	50	CP025570	complete	(this study)
*A. jensenii* NCTC13652 ^T^	3180547	68.5	2839	2684	4,4,4	50	NZ_LR134473.1	complete	N/A *
*A. thoenii* DSM 20276 ^T^	2938072	68	2678	2617	3,4,1 **	50	NZ_KE384018.1	draft	N/A
*A. acidipropionici* CGMCC1.2230 ^T^	3651382	68.8	3318	3162	4,4,4	53	NZ_CP013126.1	complete	[49]
*A. virtanenii* JS278 ^T^	3432872	68.4	3152	3086	3,3,3	56	CP025198	complete	[35]
*P. freudenreichii* DSM 20271 ^T^	2649166	67.3	2333	2280	2,2,2	44	NZ_CP010341.1	complete	[36]
*C. acnes* KPA 171202	2560265	60	2565	2416	3,3,3	45	AE017283	complete	[50]

* N/A- not available; ** The rRNA numbers are not final as this genome is at the draft level; ^T^ indicates type strain of the species.

**Table 3 microorganisms-07-00512-t003:** Phenotypic comparison of strains JS279 and JS280 with type strains of other species from the genus *Acidipropionibacterium*. (1) *A. jensenii* JS279, (2) *A. jensenii* JS280, (3) *A. jensenii* DSM 20535^T^, (4) *A virtanenii* JS278^T^, (5) *A. acidipropionici* DSM 4900 ^T^, (6) *A. thoenii* DSM 20276 ^T^, (7) *A. microaerophilum* DSM 13435 ^T^, (8) *A. olivae* IGBL1 ^T^, (9) *A. damnosum* IGBL13 ^T^.

	1	2	3	4	5	6	7	8	9
**Source**	malted barley	malted barley	buttermilk	malted barley	dairy product	cheese	olive waste-water	spoiled green olives	spoiled green olives
**Colony Colour**	orange/red-brown	cream/yellow	cream/yellow	cream/orange >7days	cream	orange/red-brown	white	cream	white to cream
**Haemolysis**	+	−	−	+	−	+	N/A	N/A	N/A
**Cell Size (µm)**	1–2	1–2	N/A	1–5	N/A	N/A	2–3.5	1.4–4	5–30
**Catalase test**	+	+	−	+	−	+	−	+	−
**Conditions for growth**									
**Temperature (°C)**	12–37	12–42	N/A	12–37	N/A	N/A	20–45	20–42	20–42
**pH**	4.5–9	4.5–9	N/A	5–9	N/A	N/A	4.5–9.5	4–10	4.5–8
**Maximum NaCl (%)**	6.5	6.5	N/A	6.5	N/A	N/A	2	4	4
**Fermentation Profile Determined by API Test**									
**d-Arabinose**	−	−	−	+^§^	+	−	−	+	−
**l-Arabinose**	-	−	−	+	+	−	+	+	+
**d-Xylose**	−	−	−	−	−	−	−	+	−
**Sorbose**	−	−	−	−	−	−	+	−	−
**Rhamnose**	−	−	−	−	+	−	+	−	+
**Inositol**	+	+	+	+	+	−	+	N/A	N/A
**Mannitol**	+	+	+	+	+	−	+	N/A	N/A
**Sorbitol**	−	−	−	+	+	+	+	N/A	N/A
**Methyl-α-d-mannopyranoside**	−	−	−	−	−	+	−	N/A	N/A
**Methyl-α-d-Glucopyranoside**	**+** ^§^	**+**	**−**	+	+	+	+	N/A	N/A
***N*-Acetyl Glucosamine**	−	−	v	+	+	+	+	−	+
**Amygdalin**	−	−	v	v	+	−	−	−	−
**Arbutin**	−	−	−	+	+	N/A	N/A	+	−
**Esculin**	+	+	+^§^	+	+	+	−	N/A	N/A
**Salicin**	**−**	**+** ^§^	**+** ^§^	+^§^	+	N/A	N/A	N/A	N/A
**d-Cellobiose**	**−**	**−**	**+**	+	+	N/A	+	+	+
**Maltose**	+	+	+	+	+	N/A	N/A	N/A	N/A
**Lactose**	−	−	−	+^§^	+	+	−	−	+
**Melibiose**	+	+	+	+	−	−	−	−	+
**Saccharose**	+	+	+	+	+	N/A	N/A	N/A	N/A
**Trehalose**	+	+	+	+	+	N/A	N/A	N/A	N/A
**Melezitose**	+	+	+^§^	−	+	−	+	−	−
**Raffinose**	+	+	+	+	−	−	−	−	−
**Amidon (starch)**	**+** ^§^	**+**	**−**	+	+	+	+	−	−
**Glycogen**	−	−	−	−	−	−	+	−	−
**Xylitol**	**+**	**+**	**−**	+	+	N/A	N/A	N/A	N/A
**Gentiobiose**	**−**	**−**	**+**	+^§^	+	N/A	N/A	N/A	N/A
**d-Fucose**	−	−	−	−	−	−	−	N/A	N/A
**Potassium Gluconate**	V ^§^	+	+^§^	+	+	−	+	−	−
**Reference**	(this study)	(this study)	[35]	[35]	[35]	[61]	[61]	[62]	[62]

Fermentation profiles were determined by the API CHL 50 test. **+**, Positive reaction; **−**, negative reaction; v, variable between replicates; N/A, data not available. API results were recorded at 72 h and 7 days of incubation. For all of the strains tested the reactions with glycerol, erythritol, ribose, adonitol, galactose, glucose, fructose, mannose and turanose, D-arabitol, and L-arabitol were positive after 72 h incubation for all of the strains tested. ^§^ Reactions recorded as positive only after 7 days of incubation. The reactions differing between the three *A. jensenii* strains are marked in bold.

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
