# Peer review of "Red-Brown Pigmentation of Acidipropionibacterium jensenii Is Tied to Haemolytic Activity and cyl-Like Gene Cluster"

_microorganisms, 2019, doi:10.3390/microorganisms7110512_

Round 1

Reviewer 1 Report

The paper describes new genomic and phenotypic findings on strains of the new

 Acidipropionibacterium genus, hence enlarging the knowledge on this genus. The approach is clear and also the link to practical implications for industry and human health are well described. The only minor point is the fact that the paper contains quite a number of minor language imperfections. Also, on some places long and complex sentences are used. Overall, you can improve the readability of the work to end up with a nice manuscript.

Line 21: replace “ which has been tied to rhamnolipid pigment” with “ which has been tied to the rhamnolipid pigment”

Line 27: replace “ with cyl-like gene cluster” with “ with the cyl-like gene cluster”

Line 36: replace “ that is stemmed from” with ”that originates from”

Line 53: replace “as well as, production” with ” as well as the production”

Line 63: Your write “The haemolytic activity of A. jensenii and A. thoenii strains have been tied to their red-brown pigmentation [19] and subsequently to the production of the polyene pigment granadaene 64 [20].” This sentence contains only one reference, which is also very old (1971!) so this statement is not supported very well by literature. Also the statement is not elaborated: how solid is the correlation between the pigmentation and the haemolytic activity? Is the correlation 100%? How was it demonstrated? Can you elaborate on this in the paper? Or, if you cannot elaborate it because no information is available, make this more clear to the reader. As it is written now, it is a bit confusing.

Line 70: replace “eliminate red/brow -spot defects” with “eliminate red/brown-spot defects”

Line 75: replace “haemolytic activity is a virulence factor and could indicate” with “haemolytic activity is a virulence factor that can also indicate”

Line 78: replace “inhibition by whey filtrates” by “ growth inhibition by whey filtrates”

Line 79: replace “plays a similar, so far” by “plays a similar but so far”

Line 140: You used a positive control for the haemolytic activity, but did you also use a positive control for the pigmentation?

Line 141 (title): put cylG in Italic

Line 144: replace “primers’ annealing” by “primers annealing“

Line 147 and 150: include duration and g force used for centrifugation

Line 158: replace “by comparingagainst” by “by comparing against”

Line 173: replace “like previously” by “as described previously”

Line 196: Is there “***” missing before “Not type“?

Line 202: “cylJ and cylK”: and should not be italic

Line 221: “which based on”: do you mean “which was based on”? Sentence is long, complex and difficult to read. Please simplify.

Line 216: The text below the figure and above the figure legend is not readable. Include this text in the legend.

Line 229: “Clearance around bacterial growth”: was there a minimal diameter to be present for you to decide for a positive reaction, some kind of ‘minimal signal’? How much did you repeat this assay and were the results the same in all repetitions?

Line 267: “However, we previously reported that the members of the genus Acidipropionibacterium lack the bluB part of the fusion gene bluB/cobT2 necessary for the production of 5,6-dimethylbenzimidazole (DMBI), the lower ligand of the active form of vitamin B12 [35] and that the presence of aspartate in the active site of the product of cobT gene responsible for activation of the lower ligand for attachment indicates preference for activation of lower ligand bases less hydrophobic than DMBI, such as adenine, which results in a preferential production of pseudovitamin B12 [55].” This is one very very long sentence. Please improve its readability and cut into three to four sentences.

Line 294: Sentence starting with “CRISPR2 consists …”: same remark.

Line 319: replace “in strain JS279, however it is” by “in strain JS279. However, it is”

Line 339: “starch (amidon)”: why amidon between brackets? What do you want to say with this?

Line 360 to 369: While this is a very clear part in the text, my feeling is that there is too much repetition of the problem statement in the conclusions section. This part of the text belongs in the introduction, allowing you to come faster to the conclusion of YOUR study in the conclusions section. In my opinion, the conclusions section can start at line 369 with the sentence “Linking the …” and so on.

Author Response

Response to Reviewer 1 Comments

Point 1:

” The only minor point is the fact that the paper contains quite a number of minor language imperfections. Also, on some places long and complex sentences are used. Overall, you can improve the readability of the work to end up with a nice manuscript.”

Response 1:

We appreciate the thoughtful and kind comments of the Reviewer 1. The point-by-point responses are presented below, together with the comments they address.

Line 21: replace “which has been tied to rhamnolipid pigment” with “which has been tied to the rhamnolipid pigment” Replaced.

Line 27: replace “with cyl-like gene cluster” with “with the cyl-like gene cluster” Replaced.

Line 36: replace “that is stemmed from”with ”that originates from” Replaced.

Line 53: replace “as well as, production” with ”as well as the production” Replaced.

Line 63: Your write “The haemolytic activity of A. jensenii and A. thoenii strains have been tied to their red-brown pigmentation [19] and subsequently to the production of the polyene pigment granadaene 64 [20].” This sentence contains only one reference, which is also very old (1971!) so this statement is not supported very well by literature. Also the statement is not elaborated: how solid is the correlation between the pigmentation and the haemolytic activity? Is the correlation 100%? How was it demonstrated? Can you elaborate on this in the paper? Or, if you cannot elaborate it because no information is available, make this more clear to the reader. As it is written now, it is a bit confusing.

We appreciate the comment. In order to improve clarity of the sentence, we rephrased it as follows: 

The haemolytic activity of A. jensenii and A. thoenii strains has been tied to their red-brown pigmentation [19], which was later shown to result from the production of the polyene pigment granadaene [20]. “We hope that this clarifies that the red-brown colour of the haemolytic strains results from the production of red pigment granadaene. Non-pigmented strains are not haemolytic, which we demonstrate in the Results and Discussion section.

Line 70: replace “eliminate red/brow -spot defects” with “eliminate red/brown-spot defects” Replaced.

Line 75: replace “haemolytic activity is a virulence factor and could indicate” with “haemolytic activity is a virulence factor that can also indicate” Replaced.

Line 78: replace “inhibition by whey filtrates” by “growth inhibition by whey filtrates” Replaced.

Line 79: replace “plays a similar, so far” by “plays a similar but so far” Replaced.

Line 140: You used a positive control for the haemolytic activity, but did you also use a positive control for the pigmentation?

As clarified in line 63, it was previously shown that pigmentation results in haemolytic activity in Acidipropionibacterium strains. This, combined with the description of similar relationship between the pigmentation and haemolysis in S. agalactiae (lines 82-84), was deemed to sufficiently link pigmentation and haemolysis, without need for separate pigmentation positive controls in the present study.

Line 141 (title): put cylG in Italic. Format changed to Italic.

Line 144: replace “primers’ annealing” by “primers annealing“. Replaced.

Line 147 and 150: include duration and g force used for centrifugation. “at 3220 g for 10 minutes” added.

Line 158: replace “by comparingagainst” by “by comparing against”. Replaced.

Line 173: replace “like previously” by “as described previously”. Replaced.

Line 196: Is there “***” missing before “Not type“? Indeed, it was missing. “***” added.

Line 202: “cylJ and cylK”: and should not be italic. Format of “and” changed from Italic.

Line 221: “which based on”: do you mean “which was based on”? Sentence is long, complex and difficult to read. Please simplify.

The long sentence was split into two and currently reads: “The presence of a cyl gene cluster was previously reported for a genome assembled from a metagenomic sample, Propionibacterium sp. 5 U 42AFAA [26]. The sample is most likely a strain of Cutibacterium acnes subsp. acnes based on an ANIb value of 99.7% when compared to Cutibacterium acnes type strain ATCC 6919. “.

Line 216: The text below the figure and above the figure legend is not readable. Include this text in the legend. The text was removed from the figure and incorporated into the figure legend.

Line 229: “Clearance around bacterial growth”: was there a minimal diameter to be present for you to decide for a positive reaction, some kind of ‘minimal signal’? How much did you repeat this assay and were the results the same in all repetitions?

The reaction was assessed on the basis of presence/absence of clear zone around the bacterial streak as we were determining the presence of the effect rather than its scale. Measuring the size of the clearance zone would additionally require controlling for the number of bacterial cells used and even the intensity of pigmentation, which was beyond the scope of the current study. The experiments were repeated at least three times with the same result for each strain, and as explained in the legend of Figure 3, only the representative results are shown. For clarity, the information about the number of repetitions was added into the “Materials and Methods” section of the manuscript (line 140).

Line 267: “However, we previously reported that the members of the genus Acidipropionibacterium lack the bluB part of the fusion gene bluB/cobT2 necessary for the production of 5,6-dimethylbenzimidazole (DMBI), the lower ligand of the active form of vitamin B12 [35] and that the presence of aspartate in the active site of the product of cobT gene responsible for activation of the lower ligand for attachment indicates preference for activation of lower ligand bases less hydrophobic than DMBI, such as adenine, which results in a preferential production of pseudovitamin B12 [55].” This is one very very long sentence. Please improve its readability and cut into three to four sentences.

The admittedly very long and complex sentence was split into four and currently reads: “However, we previously reported that the members of the genus Acidipropionibacterium lack the bluB part of the fusion gene bluB/cobT2 necessary for the production of 5,6-dimethylbenzimidazole (DMBI), the lower ligand of the active form of vitamin B12 [35]. Also, the presence of aspartate in the active site of the product of cobT gene is decisive for its function. CobT enzyme is responsible for activation of the lower ligand prior to its attachment. The charged amino acid in the active site indicates preference for activation of lower ligand bases less hydrophobic than DMBI, such as adenine, which results in a preferential production of pseudovitamin B12 [55].”

Line 294: Sentence starting with “CRISPR2 consists …”: same remark.

This very long and complex sentence was split into three and currently reads: “CRISPR2 consists of 76 spacers located between 280463-285125 and is associated with a complete CRISPR-Cas type IE system (Figure 4a). CRISPR1 consists of two spacers and is located directly upstream of the CRISPR2 array (279029-279180). CRISPR1 and CRISPR2 are separated by three genes coding for hypothetical proteins, each with similarity to transposases.”

Line 319: replace “in strain JS279, however it is” by “in strain JS279. However, it is”. Replaced.

Line 339: “starch (amidon)”: why amidon between brackets? What do you want to say with this?

The name of the carbon source was presented like on the information sheet of the API CHL kit. “Amidon” part is not necessary and was removed.

Line 360 to 369: While this is a very clear part in the text, my feeling is that there is too much repetition of the problem statement in the conclusions section. This part of the text belongs in the introduction, allowing you to come faster to the conclusion of YOUR study in the conclusions section. In my opinion, the conclusions section can start at line 369 with the sentence “Linking the …” and so on.

We agree that the Conclusions section was too long (also pointed out by Reviewer 2). We therefore shortened the section as suggested, with the exception of removal of the first sentence. The reason for keeping the first sentence is to highlight the importance and health-promoting potential of A. jensenii and A. thoenii. Without those benefits, it could be argued that due to the potential risks associated with pigmented strains all strains of these species should be avoided in food applications.

We thank Reviewer 1 for the detailed and thoughtful comments. We believe that addressing them improved the quality and readability of the manuscript.

Reviewer 2 Report

This manuscript describes the characterization of strains of Acidipropionibacterium jensenii with particular emphasis on its hemolytic activity, the production of granadaene, and the presence of a cyl-like gene cluster.  The research appears well done and quite detailed in their characterization of the strains they examined, and ultimately appropriate for publication after the authors deal with several editorial issues.  These are outlined below.

Line 40.  Is “stemmed” the correct word to describe the reexamination of the classification of a genus and the subsequent splitting off of a new genus?

Linee 44.  Delete “also”.

Introduction and elsewhere in the manuscript.  The authors have a propensity to use “run-on” that become complex and not always easy to interpret.  The authors should review and break up such sentences into two or more simpler sentences.

Line 70.  “red/brown-spot” needs to be corrected.

Lines 139-140.  Interestingly, the S. aureus used as a control for B-hemolytic activity is capable of producing a “golden-yellow” carotenoid pigment that has been linked to its virulence as a pathogen.  Did the authors by any chance examine the S. aureus isolate for any overlap with the cyl-like activities in A. jensenii?

Line 158.  “comparing against” 

Line 263.  Do you mean “samples” or should it be “strains?”

Line 267.  Traditionally the 12 in vitamin B12 is a subscript.

Table 3.  Any idea whether the pathways from the various carbohydrates to granadaene generate NADH or NADPH, and which is required for the pigment synthesis?

Author Response

Response to Reviewer 2 Comments

Point 1:

This manuscript describes the characterization of strains of Acidipropionibacterium jensenii with particular emphasis on its hemolytic activity, the production of granadaene, and the presence of a cyl-like gene cluster.  The research appears well done and quite detailed in their characterization of the strains they examined, and ultimately appropriate for publication after the authors deal with several editorial issues.  These are outlined below.

Response 1:

We appreciate the thoughtful comments of the Reviewer 2. The point-by-point responses are presented below, together with the comments they address.

Line 40.  Is “stemmed” the correct word to describe the reexamination of the classification of a genus and the subsequent splitting off of a new genus? “Stemmed” has been replaced by “originates”.

Linee 44.  Delete “also”. Deleted.

Introduction and elsewhere in the manuscript.  The authors have a propensity to use “run-on” that become complex and not always easy to interpret.  The authors should review and break up such sentences into two or more simpler sentences.

We recognise that the long sentences make the article difficult to follow. In order to improve readability, multiple sentences were broken down. As a rule, every sentence extending over three lines of text was split.

As an example: “In the pathogenic bacterium Streptococcus agalactiae, the production of granadaene and the associated haemolytic activity is an important virulence factor, implicated in the bacterium's ability to efficiently infect human hosts, [25] and was recently shown to enable penetration of the amniotic cavity in pregnant women, which is associated with preterm births and foetal injury [26].” now reads: “In the pathogenic bacterium Streptococcus agalactiae, the production of granadaene and the associated haemolytic activity is an important virulence factor. It is implicated in the bacterium's ability to efficiently infect human hosts, [25] and was recently shown to enable penetration of the amniotic cavity in pregnant women, which is associated with preterm births and foetal injury [26].” also“However, we previously reported that the members of the genus Acidipropionibacterium lack the bluB part of the fusion gene bluB/cobT2 necessary for the production of 5,6-dimethylbenzimidazole (DMBI), the lower ligand of the active form of vitamin B12 [35] and that the presence of aspartate in the active site of the product of cobT gene responsible for activation of the lower ligand for attachment indicates preference for activation of lower ligand bases less hydrophobic than DMBI, such as adenine, which results in a preferential production of pseudovitamin B12 [55].” now reads: “However, we previously reported that the members of the genus Acidipropionibacterium lack the bluB part of the fusion gene bluB/cobT2 necessary for the production of 5,6-dimethylbenzimidazole (DMBI), the lower ligand of the active form of vitamin B12 [35]. Also, the presence of aspartate in the active site of the product of cobT gene is decisive for its function. CobT enzyme is responsible for activation of the lower ligand prior to its attachment. The charged amino acid in the active site indicates preference for activation of lower ligand bases less hydrophobic than DMBI, such as adenine, which results in a preferential production of pseudovitamin B12 [55].”

Line 70.  “red/brown-spot” needs to be corrected. Corrected.

Lines 139-140.  Interestingly, the S. aureus used as a control for B-hemolytic activity is capable of producing a “golden-yellow” carotenoid pigment that has been linked to its virulence as a pathogen.  Did the authors by any chance examine the S. aureus isolate for any overlap with the cyl-like activities in A. jensenii?

This is a very interesting question. Although the reports in literature indicate that the pigment of S. aureus is a carotenoid without a direct role in haemolysis a potential overlap in activity cannot be excluded. However, we have not yet studied cyl-like activities in S. aureus.

Line 158.  “comparing against” . Words were separated.

Line 263.  Do you mean “samples” or should it be “strains?” Changed to “strains”.

Line 267.  Traditionally the 12 in vitamin B12 is a subscript. “12” has been changed to subscript.

Table 3.  Any idea whether the pathways from the various carbohydrates to granadaene generate NADH or NADPH, and which is required for the pigment synthesis? Again, we thank the referee for a very interesting question. However, we have not explored the biosynthetic pathway of granadaene in detail. The ability of strains JS278 and JS279 to utilise many of the same carbon sources and having pigment-production patterns differing considerably suggest complex relationship between pigment production and the carbon source, which require further study.

We thank Reviewer 2 for interesting and thought-provoking comments. We believe that addressing them improved the quality and readability of the manuscript.

Reviewer 3 Report

The manuscript presented for review under the title "Red-brown pigmentation of Acidipropionibacterium jensenii is tied to haemolytic activity and cyl-like gene cluster" is valuable. The authors demonstrated, that PCR detection with cylG as a target shows diagnostic potential in detection of haemolytic and pigmented strains among several Acidipropionbacterium species. And also: i) they reported whole genome sequences of 2 novel strains of A. jensenii (pigmented and non-pigmented phenotype), ii) they compared the sequenced genomes to those defined for the type strain A. jensenii and other acidipropionibacteria, iii) they complemented genomic findings related to B12 vitamin synthesis, genome integrity, haemolytic activity and carbohydrate fermentation profiles with necessary bioinformatic and phenotypic analyses.

The manuscript is prepared very carefully, minor remarks are included in the attached pdf file (they refer mainly to: 1) supplementing in the "Materials and Methods" chapter with information about the manufacturers of the equipment or materials used, and 2) too large a font in the tables). The only major remark is: the "Conclusions" chapter is too extensive, is more like "Summary". I suggest shortening.

All corrections are marked in the attached pdf file.

Author Response

We thank Reviewer 3 for the helpful comments. For the point-by-point responses, please see the attachment.
